# Vaccine effectiveness against SARS-CoV-2 infection, hospitalization, and death when combining a first dose ChAdOx1 vaccine with a subsequent mRNA vaccine in Denmark: A nationwide population-based cohort study

**Mie Agermose Gram**[1]*, **Jens Nielsen**[1], **Astrid Blicher Schelde**[1], **Katrine Finderup Nielsen**[1], **Ida Rask Moustsen-Helms**[1], **Anne Katrine Bjørkholt Sørensen**[2], **Palle Valentiner-Branth**[1], **Hanne-Dorthe Emborg**[1]

1 Department of Infectious Disease Epidemiology and Prevention, Statens Serum Institut, Copenhagen, Denmark, 2 Department of Infectious Disease Preparedness, Data Integration and Analysis, Statens Serum Institut, Copenhagen, Denmark

* miag@ssi.dk

## Abstract

### Background

The recommendations in several countries to stop using the ChAdOx1 vaccine has led to vaccine programs combining different Coronavirus Disease 2019 (COVID-19) vaccine types, which necessitates knowledge on vaccine effectiveness (VE) of heterologous vaccine schedules. The aim of this Danish nationwide population-based cohort study was therefore to estimate the VE against Severe Acute Respiratory Syndrome Coronavirus 2 (SARS-CoV-2) infection and COVID-19–related hospitalization and death following the first dose of the ChAdOx1 vaccine and the combination of the ChAdOx1/mRNA vaccines.

### Methods and findings

All individuals alive in or immigrating to Denmark from 9 February 2021 to 23 June 2021 were identified in the Danish Civil Registration System. Information on exposure, outcomes, and covariates was obtained from Danish national registries. Poisson and Cox regression models were used to calculate crude and adjusted VE, respectively, along with 95% confidence intervals (CIs) against SARS-CoV-2 infection and COVID-19–related hospitalization or death comparing vaccinated versus unvaccinated individuals. The VE estimates were adjusted for calendar time as underlying time and for sex, age, comorbidity, country of origin, and hospital admission. The analyses included 5,542,079 individuals (97.6% of the total Danish population). A total of 144,360 individuals were vaccinated with the ChAdOx1 vaccine as the first dose, and of these, 136,551 individuals received an mRNA vaccine as the second dose. A total of 1,691,464 person-years and 83,034 SARS-CoV-2 infections were included. The individuals vaccinated with the first dose of the ChAdOx1 vaccine dose had a median age of 45 years. The study population was characterized by an equal distribution of

**Data Availability Statement:** Data cannot be shared publicly because of data protection regulation. Data are available from the Danish Health Data Authority for researchers who meet the criteria for access to confidential data. The data are available for research upon reasonable request and with permission from the Danish Data Protection Agency and the Danish Health Data Authority: https://sundhedsdatastyrelsen.dk/da/english/health_data_and_registers/research_services.

**Funding:** The authors received no specific funding for this work.

**Competing interests:** The authors have declared that no competing interests exist.

**Abbreviations:** CI, confidence interval; COVID-19, Coronavirus Disease 2019; CRS, Civil Registration System; DNPR, Danish National Patient Registry; HR, hazard ratio; ICD-10, International Classification of Diseases, 10th revision; RT-PCR, reverse transcription polymerase chain reaction; SARS-CoV-2, Severe Acute Respiratory Syndrome Coronavirus 2; VE, vaccine effectiveness.

males and females; 6.7% and 9.2% originated from high-income and other countries, respectively. The VE against SARS-CoV-2 infection when combining the ChAdOx1 and an mRNA vaccine was 88% (95% CI: 83; 92) 14 days after the second dose and onwards. There were no COVID-19–related hospitalizations or deaths among the individuals vaccinated with the combined vaccine schedule during the study period. Study limitations including unmeasured confounders such as risk behavior and increasing overall vaccine coverage in the general population creating herd immunity are important to take into consideration when interpreting the results.

## Conclusions

In this study, we observed a large reduction in the risk of SARS-CoV-2 infection when combining the ChAdOx1 and an mRNA vaccine, compared with unvaccinated individuals.

## Author summary

### Why was this study done?

- Coronavirus Disease 2019 (COVID-19) vaccination is one of the main strategies to control the COVID-19 pandemic and to avoid hospitalizations and deaths from COVID-19.

- Knowledge about the effectiveness of the combination of the ChAdOx1 and an mRNA vaccine is important due to changed recommendations regarding the use of the ChAdOx1 vaccine.

### What did the researchers do and find?

- The vaccine effectiveness (VE) was estimated using national registry data on all Danish residents during the study period (9 February to 23 June 2021).

- Vaccination with the combination of ChAdOx1 and an mRNA vaccine was associated with estimated protection of 88% against Severe Acute Respiratory Syndrome Coronavirus 2 (SARS-CoV-2) infection.

### What do these findings mean?

- The protection of vaccination with the combination of ChAdOx1 and an mRNA vaccine is similar to the findings in previous studies of homologous vaccine schedules.

- Further research with longer follow-up time is needed to confirm vaccine-induced protection against severe events, such as COVID-19–related hospitalization and death.

## Introduction

The Coronavirus Disease 2019 (COVID-19) pandemic caused by the outbreak of the novel coronavirus Severe Acute Respiratory Syndrome Coronavirus 2 (SARS-CoV-2) in Wuhan, China, has caused major global public health concerns. As of 26 July 2021, more than 192 million cases and 4.1 million deaths have been reported worldwide [1]. In Denmark, the COVID-19 vaccination program started on 27 December 2020, with the BNT162b2 mRNA vaccine from Pfizer/BioNTech, also called Comirnaty. The mRNA-1273 vaccine (SpikeVax from Moderna) and the viral vector ChAdOx1 vaccine (Vaxzevria from Oxford/AstraZeneca) were introduced in the vaccination program on 14 January and 9 February 2021, respectively. The ChAdOx1 vaccine was primarily given to frontline personnel in healthcare, elderly care, and selected parts of the social sector who were at particular risk of infection with SARS-CoV-2 or who had been identified as performing critical functions in society (referred to as frontline personnel) [2]. On 11 March 2021, it was decided to pause the use of the ChAdOx1 vaccine in the Danish COVID-19 vaccination program due to a possible link between the vaccine and very rare cases of unusual blood clots, bleeding, and low blood platelet counts [3,4]. On 14 April 2021, it was decided not to continue with the use of the ChAdOx1 vaccine in the Danish vaccination program, due to the present epidemiological situation with a relatively low SARS-CoV-2 infection rate. The Danish Health Authority announced on 16 April 2021 that individuals who had only received the first dose of the ChAdOx1 vaccine should be offered a second dose of either the BNT162b2 mRNA or the mRNA-1273 vaccine (ChAdOx1/mRNA vaccine schedule) [4]. Studies from the UK have reported vaccine effectiveness (VE) estimates between 22% and 94% following the administration of 1 dose of ChAdOx1 [5–7]. Due to changing recommendations regarding the use of the ChAdOx1 vaccine [4] and to avoid vaccine shortages, some countries are combining vaccine types [8]. This creates a need for studies of VE for heterologous vaccination schedules [9]. Immunological data on a heterologous vaccination schedule indicate that the combination of the ChAdOx1/mRNA vaccines is at least as immunogenic and protective as homologous BNT162b2 vaccination [10,11]. However, to the best of our knowledge, no previous studies have reported VE of a ChAdOx1/mRNA vaccine schedule. This study aimed to estimate VE against SARS-CoV-2 infection and COVID-19–related hospitalization and death of (1) one dose of the ChAdOx1 vaccine and (2) the ChAdOx1/mRNA vaccine schedule both compared with unvaccinated individuals.

## Methods

### Study design and population

All residents in Denmark are registered in the Danish Civil Registration System (CRS) and assigned a unique personal identification number (CPR number), which is used in all national registries, enabling accurate individual-level linkages between registries [12]. The high quality of this registry is ensured by the ongoing correction of errors and the validation of information recorded [12]. In this nationwide retrospective population-based cohort study, individuals were included in the study population if they were residents in Denmark on 9 February 2021 or immigrated before the end of study on 23 June 2021. Individuals who had received a COVID-19 vaccine or had a reverse transcription polymerase chain reaction (RT-PCR) confirmed SARS-CoV-2 infection before the start of the study (2.4%) were excluded. The latter exclusion criteria was applied due to expected natural immunity from previous SARS-CoV-2 infection [13]. All study participants were followed from 9 February 2021 and until a SARS-CoV-2 infection, receiving the first dose of another COVID-19 vaccine than the ChAdOx1 vaccine, receiving the second dose of the ChAdox1 vaccine, emigration, death, or end of

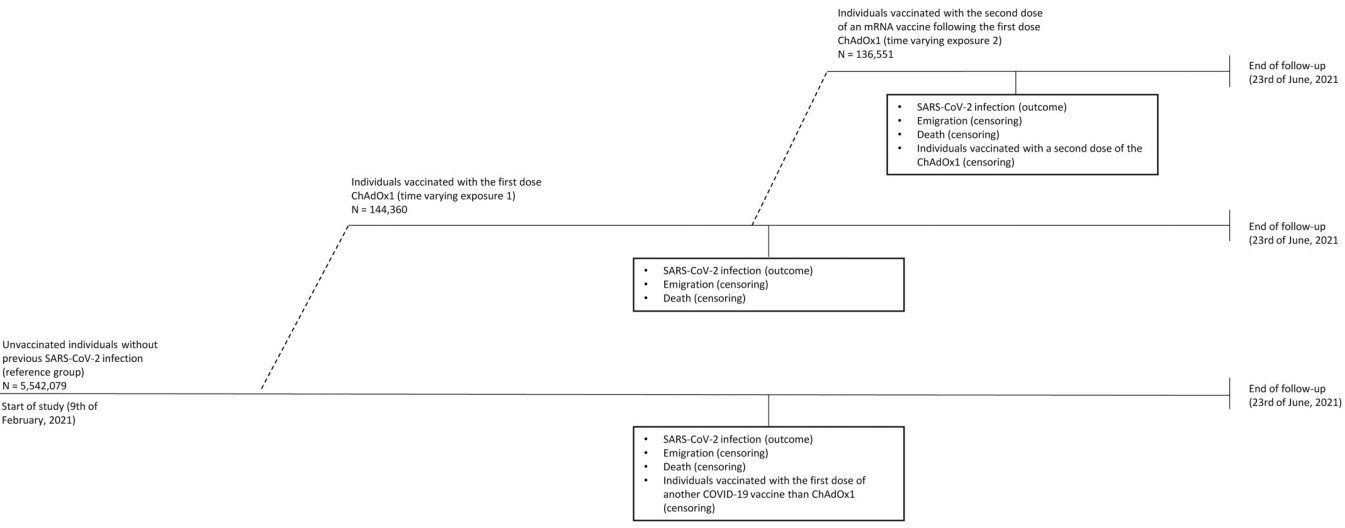

**Fig 1. Flow chart of the study population.** COVID-19, Coronavirus Disease 2019; COVID-19, Coronavirus Disease 2019.

follow-up (23 June 2021), whichever came first. The study participants contributed with follow-up time to the first exposure when they received the first dose of the ChAdOx1 vaccine and to the second exposure when they received the second dose of an mRNA vaccine following the first dose of ChAdOx1 during the study period (Fig 1). Information on immigration, emigration, and vital status was retrieved from the CRS [12]. Information on the date of vaccination and type of vaccine was retrieved from the Danish Vaccination Registry, which includes data on all administered vaccines [14]. Information on the second dose is only included if a person has completed the vaccination schedule with the correct time window relevant for the specific type of vaccine. The study did not have a prespecified analysis plan. This study is reported as per the REporting of studies Conducted using Observational Routinely-collected health Data (RECORD) Statement (S1 Checklist).

## Assessment of exposures

The exposures of interest were (1) a single dose of ChAdOx1 and (2) a second dose of either BNT162b2 mRNA or mRNA-1273 following the first dose of ChAdOx1. Unvaccinated individuals with no history of SARS-CoV-2 infection were used as reference. To examine the effect of 1 dose of ChAdOx1 on SARS-CoV-2 infection, time following vaccination was divided into 0 to 13 days (the run-in period), and 14 days and onward was divided into 7-day intervals until receiving the second dose. The effect of 1 dose of ChAdOx1 on COVID-19–related hospitalization and death was divided into 0 to 13 days (the run-in period) and 14 days and onwards until receiving the second dose of an mRNA vaccine. The effect of the combined vaccine schedule on all outcomes was divided into 0 to 13 days and 14 days and onwards.

## Assessment of outcomes

The outcomes of interest were a SARS-CoV-2 infection, defined as a laboratory-confirmed RT-PCR SARS-CoV-2–positive test, and COVID-19–related hospitalization and death. Information on RT-PCR SARS-CoV-2–positive tests was retrieved from the Danish Microbiology Database (MiBa), which is a national database that automatically accumulates both positive and negative test results from all Danish departments of clinical microbiology [15]. Information about rapid antigen tests was not included due to moderate sensitivity in asymptomatic

patients compared with RT-PCR [16]. A COVID-19–related hospitalization or death was defined as an admission within 14 days or death within 30 days after a positive SARS-CoV-2 test, respectively. Therefore, time at risk for the COVID-19–related outcomes were extended to 14 and 30 days after a confirmed SARS-CoV-2 infection, respectively. Hospital admissions and discharge dates were retrieved from the Danish National Patient Registry (DNPR) [17].

## Covariates

The incidence of SARS-CoV-2 varied considerably throughout the study period. To control for this variation through the study period, calendar time was used as the underlying time. Further, age, sex, country of origin, and comorbidity were also included as covariates for the association between COVID-19 vaccination and SARS-CoV-2 infection, hospitalization, and death. Finally, hospital admission was included as a covariate for the association between COVID-19 vaccination and SARS-CoV-2 infection and death. Information on date of birth and sex (male/female) was retrieved from the CRS [12]. The presence of comorbidity (yes/no) within the previous 5 years (data retrieved at start of study) was identified based on diagnoses coded according to the International Classification of Diseases, 10th revision (ICD-10). Diagnoses included 1 primary and optional secondary diagnosis for each patient contact and were retrieved from the DNPR [17]. The ICD-10 codes included in the comorbidity covariate are provided in S1 Table. Information on country of origin (Denmark/high-income/other/unknown) was retrieved from the CRS (S2 Table) [12]. Only the country of origin variable contained missing observations (0.2%); these were included as unknown.

## Statistical analysis

Characteristics of the included population were described using proportions. Crude VE estimates were calculated using a Poisson regression with overdispersion (quasi-Poisson) and time at risk as offset. Adjusted hazard ratios (HRs) were obtained using a Cox regression model with calendar time as underlying time due to varying incidence of SARS-CoV-2 during the study period. Sex, country of origin, and comorbidity were included as fixed covariates, being admitted to hospital as a time varying covariate, and age as a cubic spline. Time intervals after vaccination were included as a time varying covariate. The VE estimates were calculated as (1−HR)·100%. The assumption of proportionality of hazards was assessed graphically and found to be valid.

A sensitivity analysis of the VE against SARS-CoV-2 infection following the ChAdOx1/BNT162b2 mRNA vaccine schedule and the ChAdOx1/mRNA-1273 vaccine schedule, respectively, was conducted to assess if there was any difference in VE between the BNT162b2 mRNA and the mRNA-1273 vaccine as the second dose.

Data were analyzed using R version 4.0.5 (R Foundation for Statistical Computing; https://www.R-project.org/).

## Ethical considerations

The present study was based on existing administrative data and did not require ethical approval.

## Results

The study population included 5,542,079 individuals, among which 144,360 individuals (2.6%) received the ChAdOx1 vaccine as the first dose. Of these, 88,050 (61%) and 48,501 (33.6%) individuals received the BNT162b2 mRNA and the mRNA-1273 vaccine as the second dose,

respectively. A total of 1,691,464 person-years and 83,034 SARS-CoV-2 infections were included. The individuals vaccinated with the first dose of the ChAdOx1 vaccine dose had a median age of 45 years. In the total study population, 25.6% had at least one of the comorbidities defined in S1 Table. The comorbidity category other diseases, cardiovascular diseases, and respiratory diseases were most prevalent. Furthermore, 6.7% and 9.2% originated from high-income and other countries, respectively, and there was an equal distribution of males and females (Table 1). However, the proportion of females were higher among the individuals who received the first dose of the ChAdOx1 vaccine (79.8%).

The incidence of SARS-CoV-2 infection decreased from 27 December 2020 where the vaccination program was initiated in Denmark (Fig 2). Most likely, the decrease was a result of the combination of the newly started vaccination program and the lockdown from 16 December 2020 until 1 March 2021. The partial reopening in March 2021 resulted in a slight increase in SARS-CoV-2 infections until 1 June 2021, where approximately 35% and 20% of the population had started or completed vaccination, respectively (Fig 2). As of 26 July 2021, 74.3% of all SARS-CoV-2 RT-PCR–positive tests registered in 2021 had been sequenced (Fig 2). Based on the available sequencing data, the B.1.1.7 variant (Alpha) was observed throughout the whole study and was dominant after mid-February. The B.1.617.2 variant first became prominent after the end of the present study (23 June 2021), and during the study period, only a small proportion of this variant was observed (Fig 2).

The adjusted VE estimates against SARS-CoV-2 infection between 14 and 83 days after 1 dose of the ChAdOx1 vaccine were relatively stable with overlapping confidence intervals (CIs) (Fig 3). These VE estimates ranged from 29% (95% CI: 12; 43) to 44% (95% CI: 32; 55) after 1 dose of the ChAdOx1 vaccine (Table 2). The VE estimates at 84 days and onwards were

**Table 1. Characteristics of the 5,542,079 individuals in the study population.**

| | | N | % |
|---|---|---|---|
| Number of individuals included in the analyses | | 5,542,079 | 100% |
| Median age (years) at first dose of the ChAdOx1 vaccine (IQR) | | 45 (33; 55) | |
| Median age (years) at second dose of an mRNA vaccine (IQR) | | 46 (34; 55) | |
| Sex | Male | 2,801,776 | 50.6 |
| | Female | 2,740,303 | 49.4 |
| Presence of at least 1 comorbidity | Yes | 1,416,332 | 25.6 |
| | No | 4,125,747 | 74.4 |
| Country of origin | Denmark | 4,653,808 | 84.0 |
| | High-income countries | 369,255 | 6.7 |
| | Other countries | 509,330 | 9.2 |
| | Unknown | 9,686 | 0.2 |
| Coverage of 1 dose of the ChAdOx1 vaccine | Frontline personnel | 143,231 | 2.6 |
| | Other | 1,129 | 0.02 |
| Coverage of second dose of the BNT162b2 mRNA following a first dose ChAdOx1 | | 88,050 | 1.6 |
| Coverage of the second dose of the mRNA-1273 following a first dose ChAdOx1 | | 48,501 | 0.9 |
| Median follow-up days (min, IQR, max) | | 133 (1, 97, 135, 135) | |
| Median number of days to the first dose (IQR) | | 18 (11, 24) | |
| Median number of days from the first dose to the second dose (IQR) | | 82 (78, 85) | |

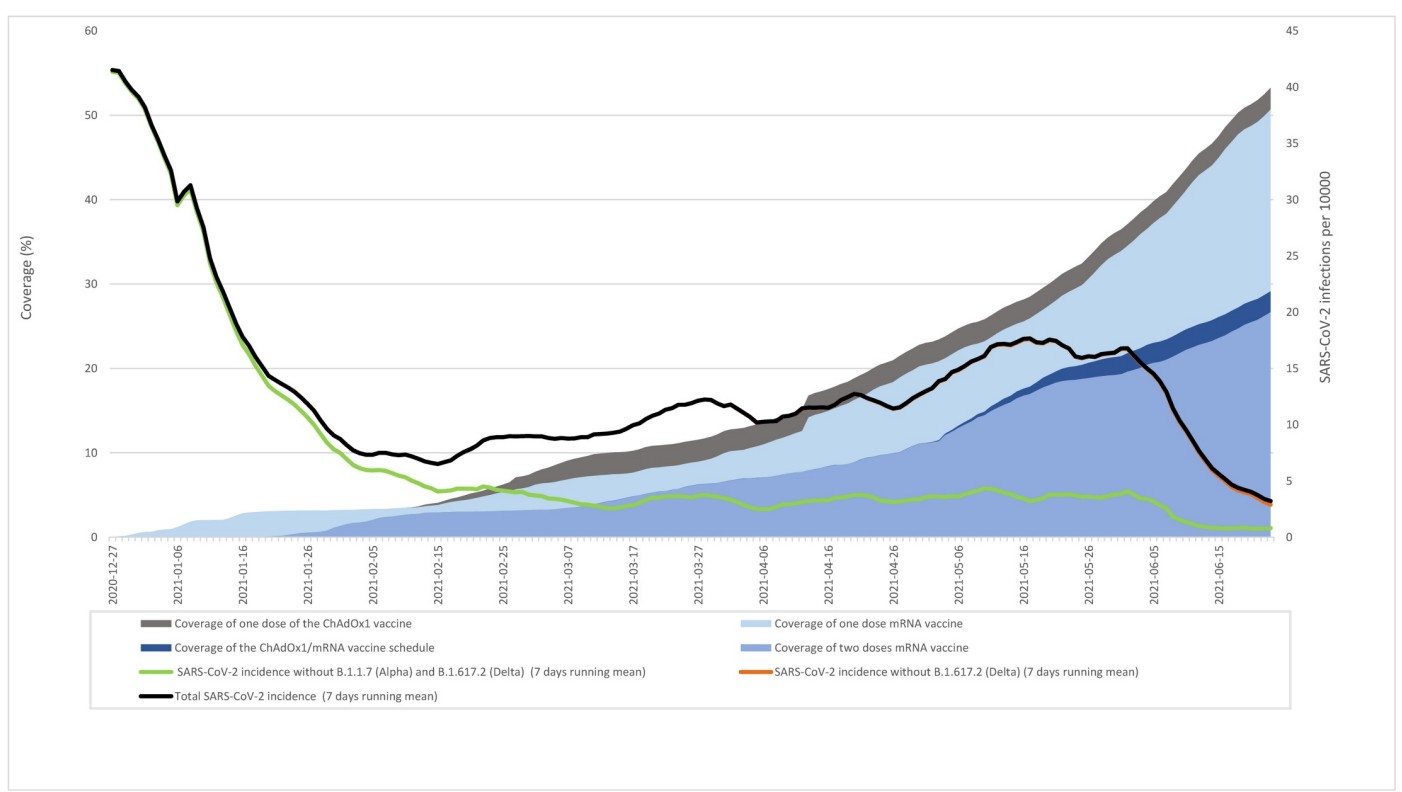

**Fig 2. Percentage of population vaccinated and incidence of SARS-CoV-2 infection (7 days running mean).** SARS-CoV-2, Severe Acute Respiratory Syndrome Coronavirus 2.

not statistically significant. However, the estimates are broadly similar throughout. The reason these VE estimates fail to meet the level of statistical significance is the increase in variability presumably related to the number of people remaining at risk at those time points. For the ChAdOx1/mRNA vaccine schedule, statistically significant adjusted VE estimates of 66% (95% CI: 59; 72) and 88% (95% CI: 83; 92) were observed at 0 to 13 days and from 14 days and onwards after the second dose, respectively (Table 2). The sensitivity analysis examining the VE against SARS-CoV-2 infection following the ChAdOx1/BNT162b2 mRNA and the ChAdOx1/mRNA-1273 vaccine schedules showed similar results (S3 Table).

A statistically significant adjusted VE of 93% (95% CI: 80; 98) against COVID-19–related hospitalization was observed from 14 days after the first dose ChAdOx1 and until receiving a second dose of an mRNA vaccine. No COVID-19–related hospitalizations occurred after the ChAdOx1/mRNA vaccine schedule. Therefore, it was not possible to estimate VE estimates against COVID-19–related hospitalization for the ChAdOx1/mRNA vaccine schedule (Table 3). Also, no COVID-19–related deaths were observed in the period after receiving the first dose of the ChAdOx1 or in the time period following the ChAdOx1/mRNA vaccine schedule (Table 3).

## Discussion

This nationwide population-based cohort study showed a significant reduction in the risk of SARS-CoV-2 infection 14 days after the second dose with a VE of 88% (95% CI: 83; 92) when combining the viral vector vaccine ChAdOx1 and an mRNA vaccine. No COVID-19–related hospitalizations or deaths occurred among individuals who received the ChAdOx1/mRNA

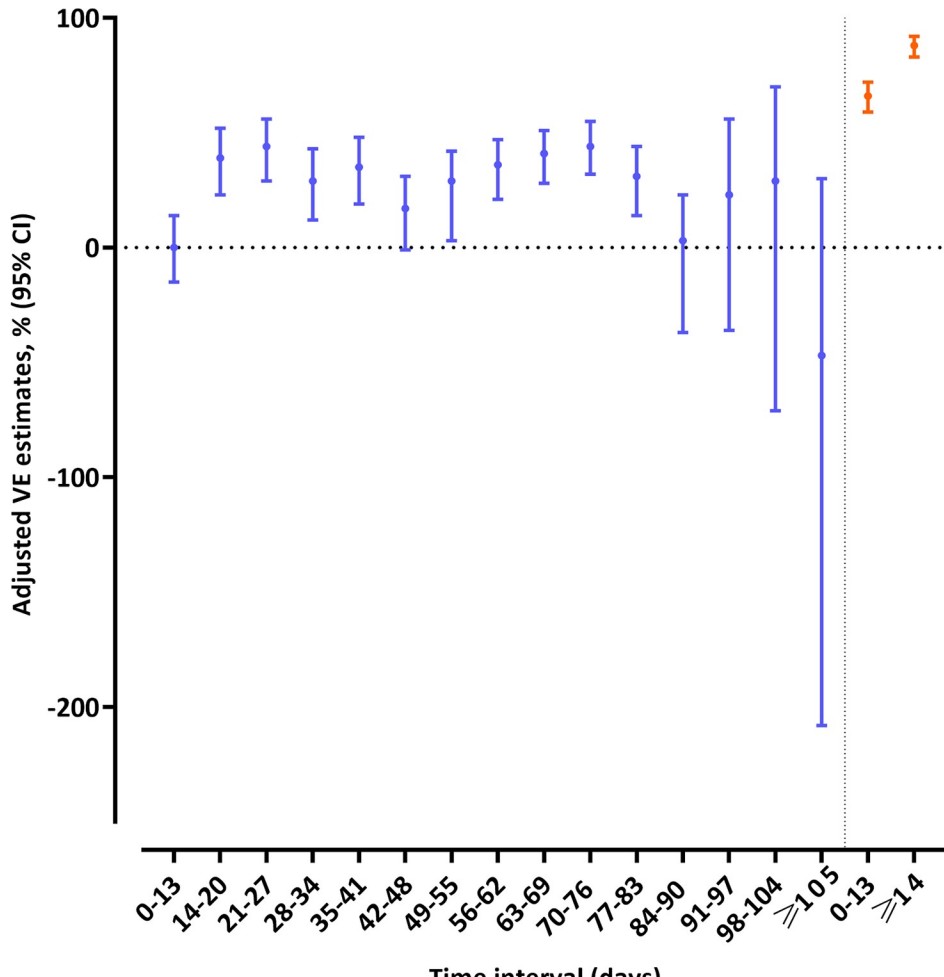

**Fig 3. Adjusted VE estimates against RT-PCR SARS-CoV-2 infection of the first dose of the ChAdOx1 vaccine and the ChAdOx1/mRNA vaccine schedule.** Each dot illustrates the estimated VE and the bars show the accompanying 95% CI. CI, confidence interval; RT-PCR, reverse transcription polymerase chain reaction; SARS-CoV-2, Severe Acute Respiratory Syndrome Coronavirus 2; VE, vaccine effectiveness.

vaccine schedule during the studied period indicating that a combined ChAdOx1 /mRNA vaccine schedule protects against severe outcomes. However, studies with longer follow-up time are needed to confirm these findings, especially because death is a rare outcome in the population of working-age.

The VE of the ChAdOx1/mRNA vaccine schedule is similar to the VE estimates of 80% (95% CI: 77; 83) and 90% (95% CI: 82; 95) reported in 2 previous Danish studies of frontline personnel (healthcare workers) who received 2 doses of the BNT162b2 mRNA vaccine [18,19]. Immunological studies also indicate that the ChAdOx1/BNT162b2 vaccine schedule is associated with stronger humoral immune responses and stronger anti-SARS-CoV-2 spike T cell responses compared with 2 doses of the ChAdOx1 vaccine [20]. Also, a 14-day robust humoral and cellular immune response after the second dose of BNT162b2 was observed in individuals primed with ChAdOx1 8 to 12 weeks earlier [21]. Heterologous vaccination schedules studies are important, as several countries want to implement a combined vaccination program with the ChAdOx1 vaccine and an mRNA vaccine to avoid restarting the vaccination schedule with 2 mRNA vaccines [8]. This is due to both changes in recommendations regarding the use of

**Table 2. Unadjusted and adjusted VE estimates against RT-PCR SARS-CoV-2 infection of 1 dose of the ChAdOx1 vaccine and the ChAdOx1/mRNA vaccine schedule.**

| Vaccine type | Time interval (days) | No. of events | Person-years | Incidence rate | Unadjusted VE, % | Unadjusted 95% CI | Adjusted* VE, % | Adjusted* 95% CI |
|---|---|---|---|---|---|---|---|---|
| Unvaccinated | | 81,755 | 1,645,793 | 0.0497 | Reference | | Reference | |
| One dose of the ChAdOx1 vaccine | 0–13 | 197 | 5,528 | 0.0356 | 28 | 8; 44 | 0 | −15; 14 |
| | 14–20 | 69 | 2,759 | 0.0250 | 50 | 23; 67 | 39 | 23; 52 |
| | 21–27 | 66 | 2,758 | 0.0239 | 52 | 26; 69 | 44 | 29; 56 |
| | 28–34 | 85 | 2,744 | 0.0310 | 38 | 8; 58 | 29 | 12; 43 |
| | 35–41 | 81 | 2,741 | 0.0296 | 41 | 12; 60 | 35 | 19; 48 |
| | 42–48 | 106 | 2,737 | 0.0387 | 22 | −10; 45 | 17 | −1; 31 |
| | 49–55 | 96 | 2,733 | 0.0351 | 29 | −1; 51 | 29 | 3; 42 |
| | 56–62 | 98 | 2,728 | 0.0359 | 28 | −3; 49 | 36 | 21; 47 |
| | 63–69 | 99 | 2,660 | 0.0372 | 25 | −7; 47 | 41 | 28; 51 |
| | 70–76 | 91 | 2,429 | 0.0375 | 25 | −9; 48 | 44 | 32; 55 |
| | 77–83 | 83 | 1,723 | 0.0482 | 3 | −43; 34 | 31 | 14; 44 |
| | 84–90 | 46 | 615 | 0.0748 | −51 | −154; 11 | −3 | −37; 23 |
| | 91–97 | 12 | 237 | 0.0506 | −2 | −183; 63 | 23 | −36; 56 |
| | 98–104 | 5 | 146 | 0.0342 | 31 | −235; 86 | 29 | −71; 70 |
| | ≥105 | 7 | 161 | 0.0436 | 12 | −234; 77 | −47 | −208; 30 |
| The ChAdOx1/mRNA vaccine schedule | 0–13 | 109 | 5,197 | 0.0210 | 58 | 41; 70 | 66 | 59; 72 |
| | ≥14 | 29 | 7,775 | 0.0037 | 92 | 86; 96 | 88 | 83; 92 |

*Adjusted for calendar time, age, sex, country of origin, hospital admission, and comorbidity.

CI, confidence interval; RT-PCR, reverse transcription polymerase chain reaction; SARS-CoV-2, Severe Acute Respiratory Syndrome Coronavirus 2; VE, vaccine effectiveness.

the ChAdOx1 vaccine and to avoid vaccine shortage. Additionally, some countries may need to combine viral vector vaccines with mRNA vaccines to boost immunity.

Due to the decision to withdraw the ChAdOx1 vaccine from the Danish vaccination program, the second vaccine dose was postponed, and it was therefore possible to evaluate the VE

**Table 3. Unadjusted and adjusted VE estimates against COVID-19–related hospitalization and death of 1 dose of the ChAdOx1 vaccine and the ChAdOx1/mRNA vaccine schedule.**

| Vaccine type | Time interval (days) | COVID-19–related hospitalization No. of events | Person-years | Incidence rate | Unadjusted VE, % | Unadjusted 95% CI | Adjusted* VE, % | Adjusted* 95% CI | COVID-19–related death No. of events | Person-years | Incidence rate | Unadjusted VE, % | Unadjusted 95% CI | Adjusted** VE, % | Adjusted** 95% CI |
|---|---|---|---|---|---|---|---|---|---|---|---|---|---|---|---|
| Unvaccinated | | 1,821 | 1,646,552 | 0.0011 | Reference | | Reference | | 122 | 1,652,106 | 0.0001 | Reference | | Reference | |
| One dose of the ChAdOx1 vaccine | 0–13 | 6 | 5,529 | 0.0011 | 19 | −3,020; 97 | 6 | −110; 58 | 0 | 5,532 | 0 | - | - | - | - |
| | ≥14 | 3 | 27,195 | 0.0001 | 90 | −1,220; 100 | 93 | 80; 98 | 0 | 27,260 | 0 | - | - | - | - |
| The ChAdOx1/mRNA vaccine schedule | 0–13 | 0 | 5,197 | 0 | - | - | - | - | 0 | 5,201 | 0 | - | - | - | - |
| | ≥14 | 0 | 7,772 | 0 | - | - | - | - | 0 | 7,782 | 0 | - | - | - | - |

*Adjusted for calendar time, age, sex, country of origin and comorbidity.

**Adjusted for calendar time, age, sex, country of origin, comorbidity and hospital admission.

CI, confidence interval; COVID-19, Coronavirus Disease 2019; VE, vaccine effectiveness.

of 1 dose of the ChAdOx1 vaccine over a longer time. The VE estimates ranged from 29% (95% CI: 12; 43) to 44% (95% CI: 32; 55) between 14 to 83 days after 1 dose of the ChAdOx1 vaccine. No effectiveness against SARS-CoV-2 infection and COVID-19–related hospital admission during the first 0 to 13 days after 1 dose of the ChAdOx1 vaccine was observed. This was expected due to the run-in period before immunity is anticipated to occur. A test-negative case–control study from England including adults aged 70 years and older showed a VE against symptomatic PCR confirmed SARS-CoV-2 infection, of 22% (95% CI: 11; 32) from 14 to 20 days, reaching a VE of 73% (95% CI: 27; 90) at 35 days and onwards after 1 dose of the ChAdOx1 vaccine [6]. Additional protection against hospital admission was observed, showing a VE against emergency hospital admission of 80% [6]. Another cohort study from England, including long-term care facility residents aged 65 years and older, showed adjusted HR against SARS-CoV-2 infection immediately after the first dose translated to VE estimates of 49% (95% CI: 01; 74) at 0 to 6 days, 42% (95% CI: 04; 65) at 7 to 13 days, 67% (95% CI: 32; 84) at 28 to 34 days, and 68% (95% CI: 34; 85) at 35 to 48 days [7]. A national prospective cohort study from Scotland reported VE estimates against hospital admission ranging between 68% (95% CI: 61; 73) and 97% (95% CI: 63; 100) at 0 to 41 days after the first dose of the ChAdOx1 vaccine [5]. The differences in the VE estimates between these studies [5–7] and the present study may be explained by differences in the study populations. As of 30 June 2021, Denmark has the highest testing rate for SARS-CoV-2 per 100,000 individuals among European countries [22]. As a result of the high testing rate, we may have detected more SARS-CoV-2 infections in Denmark than in England and Scotland and therefore observing a lower VE of 1 dose of the ChAdOx1 vaccine. Furthermore, differences in SARS-CoV-2 variants and the background risk of COVID-19 during the study period may also affect the VE estimates. Our results indicate that the ChAdOx1/mRNA vaccine schedule are effective against the Alpha variant, which became the dominating variant during the study period.

The strengths of this study are the high-quality registers and the possibility to use the unique personal identifier to link data on all residents in Denmark. The national testing strategy during the study period, including unlimited access to free-of-charge RT-PCR tests nationwide, led to a high proportion of the population being tested, which enabled us to capture data on both asymptomatic and symptomatic infections. Another strength was the access to national data on all laboratory-confirmed RT-PCR SARS-CoV-2 infections. Also, a high sensitivity (97.1%) and specificity (99.98%) was observed for the RT-PCR test [23], ensuring a low risk of misclassification. An effort was made to ensure that all individuals had equal opportunities to receive the COVID-19 vaccines. This was done through an online booking system, special campaigns, offering vaccination in some workplaces, translating the material to other languages than Danish and English, and arranging transport for those who were not able to reach the vaccination clinics on their own.

The study also has some limitations. Since we were not able to discriminate between asymptomatic and symptomatic infections, it was not possible to assess the severity of a COVID-19 infection after vaccination based on symptoms. We used a positive SARS-CoV-2 test prior to hospitalization and death (i.e., COVID-19–related hospitalization and death) as a proxy for the severity of COVID-19, although this definition might be subject to uncertainty since COVID-19 may not be the cause of these outcomes. Another limitation is that differences in test behavior related to vaccination status may exist, which may result in capturing less asymptomatic infections in vaccinated individuals, thereby leading to elevated VE estimates. The Cox regression models were adjusted for potential confounders including calendar time, age, sex, country of origin, hospital admission, and comorbidity. However, we cannot eliminate differences in health-seeking behavior or test activity as well as residual confounding related to the dichotomous classification of the comorbidity covariate. This classification does not allow

for the exclusion of differences in individual comorbidities across vaccination status, such as vaccinated individuals being more or less burdened by comorbid conditions than unvaccinated individuals. It was mainly frontline personnel who received the ChAdOx1 vaccine (99.3%). Therefore, we cannot eliminate confounding by indication, assuming that frontline personnel are more exposed to SARS-CoV-2 than the general population [24,25], which could result in a lower VE in this group. However, frontline personnel are also better trained to use personal protective equipment than the general population, which would moderate the increased risk from their occupational setting. The results may not be generalizable to other settings with a circulation of more transmissible SARS-CoV-2 variants such as the Delta variant. Through the study period, many individuals received COVID-19 vaccines other than ChAdOx1/mRNA vaccines, thus increasing the overall vaccine coverage in the general population and thereby creating indirect protective herd immunity.

In conclusion, a high protection against SARS-CoV-2 infection was observed with the ChAdOx1/mRNA vaccine schedule. No COVID-19–related hospitalizations were observed with the combined ChAdOx1/mRNA vaccine schedule. Furthermore, no COVID-19–related deaths were observed after neither the first dose of the ChAdOx1 vaccine nor the ChAdOx1/mRNA vaccine schedule. However, studies with longer follow-up time are needed to confirm these findings, especially because death is a rare outcome in the population of working-age.

## Supporting information

**S1 Checklist. The RECORD statement—checklist of items, extended from the STROBE statement, which should be reported in observational studies using routinely collected health data.**
(DOCX)

**S1 Table. Overview of the International Classification of Diseases, 10th revision (ICD-10) codes included in the comorbidity covariate.**
(DOCX)

**S2 Table. Definition of country of origin.**
(DOCX)

**S3 Table. Unadjusted and adjusted vaccine effectiveness (VE) estimates against RT-PCR SARS-CoV-2 infection of 1 dose of the ChAdOx1 vaccine and the ChAdOx1/BNT162b2 mRNA or ChAdOx1/mRNA-1273 vaccine schedule, respectively.**
(DOCX)

## Acknowledgments

The authors are grateful to the Danish Health Data Authority for their help in defining the population. We would also like to thank the Department of Data Integration and Analysis at Statens Serum Institut for data management.

## Author Contributions

**Formal analysis:** Jens Nielsen.

**Writing – original draft:** Mie Agermose Gram.

**Writing – review & editing:** Jens Nielsen, Astrid Blicher Schelde, Katrine Finderup Nielsen, Ida Rask Moustsen-Helms, Anne Katrine Bjørkholt Sørensen, Palle Valentiner-Branth, Hanne-Dorthe Emborg.

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
