## [Editor Report · Decision Letter 0]

27 Jul 2021

Dear Dr Gram, 

Thank you for submitting your manuscript entitled "Vaccine effectiveness when combining the ChAdOx1 vaccine as the first dose with an mRNA COVID-19 vaccine as the second dose" for consideration by PLOS Medicine.

Your manuscript has now been evaluated by the PLOS Medicine editorial staff and I am writing to let you know that we would like to send your submission out for external peer review.

Kind regards,

Louise Gaynor-Brook, MBBS PhD

Senior Editor

PLOS Medicine

---

## [Decision Letter · Decision Letter 1]

6 Sep 2021

Dear Dr. Gram,

Thank you very much for submitting your manuscript "Vaccine effectiveness when combining the ChAdOx1 vaccine as the first dose with an mRNA COVID-19 vaccine as the second dose" (PMEDICINE-D-21-03254R1) for consideration at PLOS Medicine. 

[LINK]

In light of these reviews, I am afraid that we will not be able to accept the manuscript for publication in the journal in its current form, but we would like to consider a revised version that addresses the reviewers' and editors' comments. Obviously we cannot make any decision about publication until we have seen the revised manuscript and your response, and we plan to seek re-review by one or more of the reviewers. 

We expect to receive your revised manuscript by Sep 27 2021 11:59PM. Please email us (plosmedicine@plos.org) if you have any questions or concerns.

We look forward to receiving your revised manuscript. 

Sincerely,

Louise Gaynor-Brook, MBBS PhD

Associate Editor 

PLOS Medicine

plosmedicine.org

Comments from the Academic Editor: 

There is a lot unclear about the methods and statistical analysis of the study. Who was included, what was the vaccination status of the rest of the population, were there people with 2 mRNA vaccinations, would it make sense to compare these to the ones included in the study? Also, the statistical analysis is described only very briefly, without any justification for underlying assumptions. How do the estimated HRs and incidences compare to the observed incidence at the time of observation? 

I am sure that the authors have a very good and complete data set, so it would be worthwhile extracting more complete information, adding new analyses and substantially improving presentation of the methods. 

General comments:

Throughout the paper, please adapt reference call-outs to the following style: "... low blood platelet counts [3,4]." (noting the absence of spaces within the square brackets).

Data availability:

PLOS Medicine requires that the de-identified data underlying the specific results in a published article be made available, without restrictions on access, in a public repository or as Supporting Information at the time of article publication, provided it is legal and ethical to do so. If the data are not freely available, please describe briefly the ethical, legal, or contractual restriction that prevents you from sharing it. Please state the owner of the data set and contact information for data requests (web or email address). Note that a study author cannot be the contact person for the data.

Title: Please revise your title according to PLOS Medicine's style. Please place the study design in the subtitle (ie, after a colon). We suggest “Vaccine effectiveness against SARS-CoV-2 infection, hospitalization and death when combining a first-dose ChAdOx1 vaccine with a subsequent mRNA vaccine in Denmark: a nationwide population-based cohort study” or similar

Abstract:

Please structure your abstract using the PLOS Medicine headings (Background, Methods and Findings, Conclusions).

Abstract Background: Please expand upon the context of why the study is important. The final sentence should clearly state the study question.

Abstract Methods and Findings:

Please provide brief demographic details of the study population (e.g. sex, age, ethnicity, etc)

Please include the dates during which the study took place and length of follow up. 

In the last sentence of the Abstract Methods and Findings section, please describe 2-3 of the main limitations of the study's methodology

Abstract Conclusions:

Please begin your Abstract Conclusions with "In this study, we observed ..." or similar, to summarize the main findings from your study without overstating your conclusions.

Author Summary:

In the final bullet point of ‘What Do These Findings Mean?’, please describe the main limitations of the study in non-technical language.

Introduction:

Line 66 - please temper assertions of primacy by adding ‘to the best of our knowledge’ or similar.

Methods:

Did your study have a prospective protocol or analysis plan? Please state this (either way) early in the Methods section. If a prospective analysis plan (from your funding proposal, IRB or other ethics committee submission, study protocol, or other planning document written before analyzing the data) was used in designing the study, please include the relevant prospectively written document with your revised manuscript as a Supporting Information file to be published alongside your study, and cite it in the Methods section. A legend for this file should be included at the end of your manuscript. If no such document exists, please make sure that the Methods section transparently describes when analyses were planned, and if/when reported analyses differed from those that were planned. Changes in the analysis-- including those made in response to peer review comments-- should be identified as such in the Methods section of the paper, with rationale. If a reported analysis was performed based on an interesting but unanticipated pattern in the data, please be clear that the analysis was data-driven.

Please ensure that the study is reported according to the RECORD guideline, and include the completed RECORD checklist as Supporting Information. Please add the following statement, or similar, to the Methods: "This study is reported as per the REporting of studies Conducted using Observational Routinely-collected health Data (RECORD) Statement (S1 Checklist)." The RECORD guideline can be found here: https://www.equator-network.org/reporting-guidelines/record/ When completing the checklist, please use section and paragraph numbers, rather than page numbers which will likely no longer correspond to the appropriate sections after copy-editing.

Please provide an ethics statement in your Methods section.

Results: 

Line 143 - please provide a reference to Table S1 in which details of comorbidities are provided, and provide a few examples of these comorbidities in the main text

Lines 156-178 - please clarify what is meant by ' Significant adjusted VE estimates’ and ‘VE estimates were not significant’

Discussion:

Please present and organize the Discussion as follows: a short, clear summary of the article's findings; what the study adds to existing research and where and why the results may differ from previous research; strengths and limitations of the study; implications and next steps for research, clinical practice, and/or public policy; one-paragraph conclusion.

Please remove all subheadings within your Discussion e.g. Strengths and limitations

Figures:

Please provide titles and legends for all figures (including those in Supporting Information files).

Please indicate in the figure caption the meaning of the bars [and whiskers] in Figure 2. 

Tables:

Please present numerators and denominators in your tables. 

Please specify the variables controlled for in Tables 2, 3 and 4.

References:

Please ensure that journal name abbreviations match those found in the National Center for Biotechnology Information (NCBI) databases, and are appropriately formatted and capitalised.

Please also see https://journals.plos.org/plosmedicine/s/submission-guidelines#loc-references for further details on reference formatting. 

Supplementary files: 

Please provide titles and legends for each individual table in the Supporting Information.

Comments from the reviewers:

Reviewer #1: This is a very nice and important study that utilizes the power of the Danish health informatics system, and the decision to suspend the use of ChAdOx1, to assess the efficacy of the ChAdOx1/mRNA combination.

The results are important and valuable for global vaccine policy.

Points

- two different mRNA vaccines were used. Is there sufficient statistical power to assess if there is any difference between these ? They have different doses and there is a suggestion that Moderna may be more potent. 

- the time interval between first and second dose if important for immune response. This appears to vary and I wonder if a plot of 'time between doses', with median and range, may be a useful addition? 

- very minor corrections of English - eg use of double negative in last sentence. 

Reviewer #2: This paper describes the effectiveness of a single dose of ChAdOx1 vaccination as well as ChAdOx1 followed by a mRNA vaccination. The findings from this study are important as countries are considering differing vaccination schedules and ways to conserve and share doses. Strengths of the study include its population based, large sample size, and near-complete capture of key data on infections, healthcare utilization, and vaccinations.

Specific comments for the authors:

Background - lines 56-57: "due to the present epidemiological situation" - not sure what this means? Because of the blood clot data - or because of patterns of SARS-CoV-2 circulation?

Assessment of exposure - lines 89-91: Individuals who exclusively received one or two doses of mRNA vaccines were excluded from analysis, correct? 

Not sure what is meant by "western" and "non-western" heritage. Please explain further.

Results - lines 138-140: Is it correct that 7,809 individuals received one ChAdOx1 vaccine but did not go on to receive a second dose of any product? Or did some of these individuals receive a second dose of ChAdOx1? Were these individuals excluded from the analysis, or did they contribute to the one ChAdOx1 dose VE estimate? How many people in the population were unvaccinated or exclusively received mRNA vaccines? I think it's important to report the size of the unvaccinated sample since they are used as the comparison group.

Line 168: typographical error: "receptively" should be "respectively"

All-cause mortality: This seems a little problematic as an outcome in this VE analysis because it is a very non-specific outcome. Could the authors provide a little more explanation about why they chose to investigate this outcome? Did the authors believe that COVID-related deaths would be missed with their case definition (even though they report that testing was common in Denmark)? All-cause hospitalization also seems like a non-specific outcome.

COVID-19 hospitalizations and deaths based on positive PCR results prior to events - the authors note this as a limitation. Could diagnosis codes or cause of death have been incorporated to improve the specificity of the case definition?

In Table 1, it appears the authors are able to distinguish front-line personnel from other vaccine recipients. The authors note that front-line personnel may have different SARS-CoV-2 exposures and PPE use behaviors that might have impacted VE. Did the authors consider a sensitivity analysis limiting to front-line workers to see if that impacted their findings?

Table 1 - "median number of days to the first dose" - not sure what this means? Wasn't follow up time started at the date of the first dose?

Reviewer #3: See attachment

[LINK]

---

## [Decision Letter · Decision Letter 2]

16 Nov 2021

Dear Dr. Gram,

Thank you very much for re-submitting your manuscript "Vaccine effectiveness against SARS-CoV-2 infection, hospitalization and death when combining a first dose ChAdOx1 vaccine with a subsequent mRNA vaccine in Denmark: a nationwide population-based cohort study" (PMEDICINE-D-21-03254R2) for consideration at PLOS Medicine.

I have discussed the paper with our academic editor and it was also seen again by one reviewer. I am pleased to tell you that, provided the remaining editorial and production issues are fully dealt with, we expect to be able to accept the paper for publication in the journal.

[LINK]

Please let me know if you have any questions, and we look forward to receiving the revised manuscript.   

Sincerely,

Richard Turner PhD, for Louise Gaynor-Brook, MBBS PhD

rturner@plos.org

Requests from Editors:

To your data statement (submission form), please add a web address, for example, for those wishing to inquire about access to study data. 

At line 37 and any other instances in the paper, please rephrase "heritage", "Western" and "non-Western" to avoid potential stigmatization. Are you able to substitute "Danes", "European migrants", "non-European migrants" and "others", for example?

At line 41, please adapt the final sentence of the "Methods and findings" subsection of your abstract to begin "Study limitations include ..." or similar. 

At line 43, please remove the comma after "observed" and make that "large" rather than "high" (reduction). 

At line 58 (author summary) please make that "... associated with estimated protection ...".

At line 68, please use the notation "... 4.1 million".

Please state explicitly in the Methods section (main text) that "The study did not have a prespecified analysis plan." or similar, assuming this is the case.

Please remove the statement on Data Availability from the Methods section (main text). In the event of publication, this information will appear in the article metadata, via entries in the submission form. 

Throughout the text, please style reference call-outs as follows: "... platelet counts [3,4]." (noting the absence of spaces within the square brackets). 

For reference 5 and other citations to the journal, "Lancet" will suffice as the journal name. 

Please add "[Preprint]" to reference 6 and any other preprints that are cited, unless you are able to substitute the corresponding peer-reviewed publications. 

Noting reference 7, please ensure that all references have full access details. 

Thank you for including a completed RECORD checklist. Please break this out into a separate attachment, labelled "S1_RECORD_Checklist" or similar and referred to as such in your Methods section (main text). 

Comments from Reviewers:

*** Reviewer #3: 

The authors have addressed all my points and the study is much clearer now.

Michael Dewey

***

[LINK]

---

## [Editor Report · Decision Letter 3]

24 Nov 2021

Dear Dr Gram, 

On behalf of my colleagues and the Academic Editor, Dr Kretzschmar, I am pleased to inform you that we have agreed to publish your manuscript "Vaccine effectiveness against SARS-CoV-2 infection, hospitalization and death when combining a first dose ChAdOx1 vaccine with a subsequent mRNA vaccine in Denmark: a nationwide population-based cohort study" (PMEDICINE-D-21-03254R3) in PLOS Medicine.

Prior to final acceptance, please make the following minor changes:

At line 41 (abstract), we suggest adapting the text to "Study limitations including unmeasured ..."; 

The first paragraph of the Discussion (main text) should summarize the findings, and we therefore suggest incorporating a paragraph break at line 238 prior to "The VE of ...";

Reference 13 appears to be a preprint, and if so please add "[preprint]"; and

Noting reference 20, please ensure that all references have full access details.

PRESS

Sincerely, 

Richard Turner PhD, for Louise Gaynor-Brook, MBBS PhD 

rturner@plos.org